# Springing for Safe Water: Drinking Water Quality and Source Selection in Central Appalachian Communities

**Hannah Patton** [1,*]**, Leigh-Anne Krometis** [1] **and Emily Sarver** [2]

[1] Biological Systems Engineering Department, Virginia Polytechnic Institute and State University, Blacksburg, VA 24061, USA; lehenry@vt.edu

[2] Mining and Minerals Engineering Department, Virginia Polytechnic Institute and State University, Blacksburg, VA 24061, USA; esarver@vt.edu

\* Correspondence: hpatton@vt.edu

**Abstract:** Issues surrounding water infrastructure, access, and quality are well documented in the Central Appalachian region of the United States. Even in cases where residents have in-home piped point-of-use (POU) water, some rely on alternative drinking water sources for daily needs—including water collection from roadside springs. This effort aims to better understand and document spring usage in this region by identifying the factors that influence drinking water source selection and comparing household and spring water quality to Safe Drinking Water Act (SDWA) health-based and aesthetic contaminant recommendations. Households were recruited from communities surrounding known springs in three states (Kentucky, Virginia, and West Virginia). First- and second-draw, in-home POU tap water samples were collected from participating households and compared to samples collected from local springs on the same day. Samples were analyzed for fecal indicator bacteria and inorganic ions. Study participants completed surveys to document perceptions of household drinking water and typical usage. The majority of survey participants (82.6%) did not trust their home tap water due to aesthetic issues. Water quality results suggested that fecal indicator bacteria were more common in spring water, while several metallic ions were recovered in higher concentrations from household samples. These observations highlight that health risks and perceptions may be different between sources.

**Keywords:** drinking water; water quality; springs; source selection; aesthetic contaminants; health-based contaminants

## 1. Introduction

Over one billion people lack access to adequate water supplies worldwide [1,2]. When a reliable improved piped water source is inaccessible, individuals and communities must rely on alternate sources of drinking water, often untreated and unregulated, in order to meet various daily needs. This behavior has been studied across many countries in various states of development. For example, previous work has documented that some residential community members in Karachi, Pakistan rely on private, unregulated water hydrants fed by informally developed boreholes for drinking water, while others are served by commercial water tankers [3]. Reliance on alternative, unimproved water sources is also a practice in Madagascar, as Boone et al. [4] determined that 44% of rural households (n = 1641) used surface water (including rivers, lakes, ponds, and springs) as a primary water source during the dry season and 2.4% used other sources including rainwater and water vendors. A study completed in the Guangxi Province in China by Cohen et al. [5] comparing boiled and bottled water use determined that wealthier families often choose to purchase bottled water, not generally considered an improved

drinking water source, because of its convenience and the perception that this water is safer than other water sources, even when in-home piped water is available.

Despite nearly 100% reported access to improved water sources, issues of piped and potable water access relating to water quality, distribution, and equity persist in the United States [6–9]. Water challenges can be particularly acute in rural areas of the country. For example, 29% of rural Alaskans and 30% of the Navajo Nation, a territory in the southwestern United States, do not have access to in-home drinking water [9]. An examination of the most recent long-form United States census data by Gasteyer and Vaswani [10] emphasized that rural localities were particularly vulnerable to issues of in-home drinking water access. Gasteyer and Vaswani [10], determined that roughly 1.27% of rural localities with populations below 1000 people and 1.19% of rural farm populations in the United States lack an in-home water service, well above the national average of 0.64%.

Drinking water access and quality has been a historic concern for rural inhabitants of Appalachia, an approximately 531,000 square-kilometer region that roughly aligns with the Appalachian Mountains of the eastern United States, encompassing a total of 13 states [11]. Forty-two percent of the population in Appalachia is considered rural, over twice the average for the United States of 20% [11]. Essential infrastructure for water systems and utilities is limited by social, geographic, and economic challenges in some Appalachian communities [12,13]. In the Central Appalachian region in particular, more than 10% of homes in several counties are without access to in-home drinking water [14]. Despite focused efforts to extend municipal system services in Appalachian communities, access to centralized water systems is only 75%, behind the national level of 85% [15]. While a significant portion of homes without a central service do have in-home water supplied by private springs and household wells, contamination of these systems, which often do not employ treatment, is common [16,17].

However, it is important to recognize that centralized water treatment systems are not necessarily a panacea in small, rural communities, especially where the needed economic, technical and human resources for adequate operation and maintenance may be lacking. In the United States, multiple recent studies have demonstrated that centralized systems in rural areas are significantly more likely than their urban counterparts to incur violations for failure to comply with water quality regulations under the Safe Drinking Water Act (SDWA) [18,19]. The SDWA regulates all public drinking water supplies in the United States [20,21]. It establishes maximum contaminant levels (MCLs) for water constituents (e.g., As, $Cl_2$) that are considered a threat to human health, and recommends secondary maximum contaminant levels (SMCLs) for constituents (e.g., Fe, Mn) that cause aesthetic issues of color, odor, or taste. The United States Environmental Protection Agency (USEPA) also provides health reference levels (HRLs) and guidance levels (GLs) for certain drinking water constituents (e.g., Na, Sr) that may be of harm to specific populations such as elderly people or children [22]. The SDWA requires water systems to provide consumers with annual reports regarding the quality of drinking water that they are provided [20], however, it should be noted that individuals who are served by private well or spring box systems are responsible for providing their own water quality information at their own expense.

Similarly to the previously mentioned studies reflecting water source selection internationally, several studies in the Appalachian region have reported that individuals without adequate access to water of sufficient quality or quantity in their home turn to alternative sources to satisfy daily needs [6,13,14,23–25]. These alternative sources are not without their own quality and access challenges. Blakeney and Marshall [25] indicate that many residents of Letcher County, Kentucky, despite in-home access to municipal water, rely upon bottled water and/or the collection of water from family member's homes to meet household needs, because they view their tap water as aesthetically displeasing and a potential risk to their health. Purchased (bottled) water can be both expensive and burdensome to obtain [23,26] and using water from the homes of others is a time commitment that can alter an individual's daily routine [25].

Krometis et al. [6] and Swistock et al. [27] report that some residents of Central Appalachia collect a portion or all of their household drinking water from roadside "spout" springs, i.e., untreated environmental waters. This choice presents a potential health risk, as both studies regularly detected

*Escherichia coli* in spring samples at the point of collection. Krometis et al. [6] determined that 63% of spring users from five Central Appalachian states (Virginia, West Virginia, Kentucky, Tennessee, and North Carolina) who participated in a related survey collected spring water for drinking at least weekly, citing positive perception of spring water quality, home water quality concerns, and/or intermittent or no in-home water availability as primary motivators for visiting springs.

Swistock et al. [27] reported that 30% of Pennsylvania residents attending local extension programming had consumed water from a roadside spring at some point and that 12% consumed spring water every year. The residents that consumed spring water yearly cited "taste" and the perceived "natural" state of the springs as primary motivators. While both of these studies emphasized the potential health risks associated with reliance on waters with detectable fecal indicator bacteria for potable purposes, neither directly compared the quality of roadside spring water to that of in-home piped drinking water. Given reports of inadequate rural infrastructure, it is possible that in-home water available in these communities might also pose a risk of exposure to contaminants. Understanding both the perceptions driving drinking source selection, and the differences in water quality between roadside spring and in-home point-of-use (POU) drinking water quality in Central Appalachia is essential to the design of effective interventions that minimize waterborne exposure to contaminants. This information can be used to more effectively advocate for expansions in water access through infrastructure investments.

Accordingly, the current study seeks to: (1) identify motivations influencing drinking water source selection in Central Appalachia; (2) directly compare roadside spout spring and household POU water quality in Central Appalachia; and (3) compare roadside spout spring and household POU water quality in Central Appalachia to US Safe Drinking Water Act (SDWA) recommendations. Though previous studies have examined factors that may prompt reliance on environmental rather than household sources of water in developing nations, this study appears to be the first of its kind in the United States.

## 2. Materials and Methods

### 2.1. Study Area

The targeted roadside springs are located in four counties in two Central Appalachian states (Figure 1). The region is primarily forested, though there is some agricultural presence at the Virginia sites. Spring sample sites were selected based on previous work by Krometis et al. [6] that assessed water quality at the sites and identified these springs as regularly used by members of the surrounding community as a potable water source. It is important to note that some springs are located in counties outside of those where home samples were collected, as some residents travel beyond county or state lines to collect drinking water from a specific roadside spring (i.e., some residents in Martin County, Kentucky visit a spring in Mingo County, West Virginia).

All roadside springs featured water flowing from a pipe (metal, PVC, or rubber) directly emerging from the ground or mountainside. None of the roadside springs had enclosures, caps, or any form of groundwater treatment. Water at the roadside springs runs constantly, even during dry periods. These roadside springs are untreated, unregulated, and are generally not monitored in an official capacity. Attempts to contact local public health authorities regarding knowledge of these roadside springs were generally unsuccessful. Laminated signs were attached to or near each spring advertising free and confidential POU water quality testing and providing a contact email and phone number for questions. Participants either responded directly to this advertisement, or after discussion with a neighbor or community liaison. All project recruitment and procedures were approved by the Virginia Tech Institutional Review Board (IRB #18-269). A total of 24 suites of household samples were collected from homes in four counties in three states: Martin County, Kentucky; McDowell County, West Virginia; and Craig and Floyd counties in Virginia, and compared with samples from the spring identified as their preference.

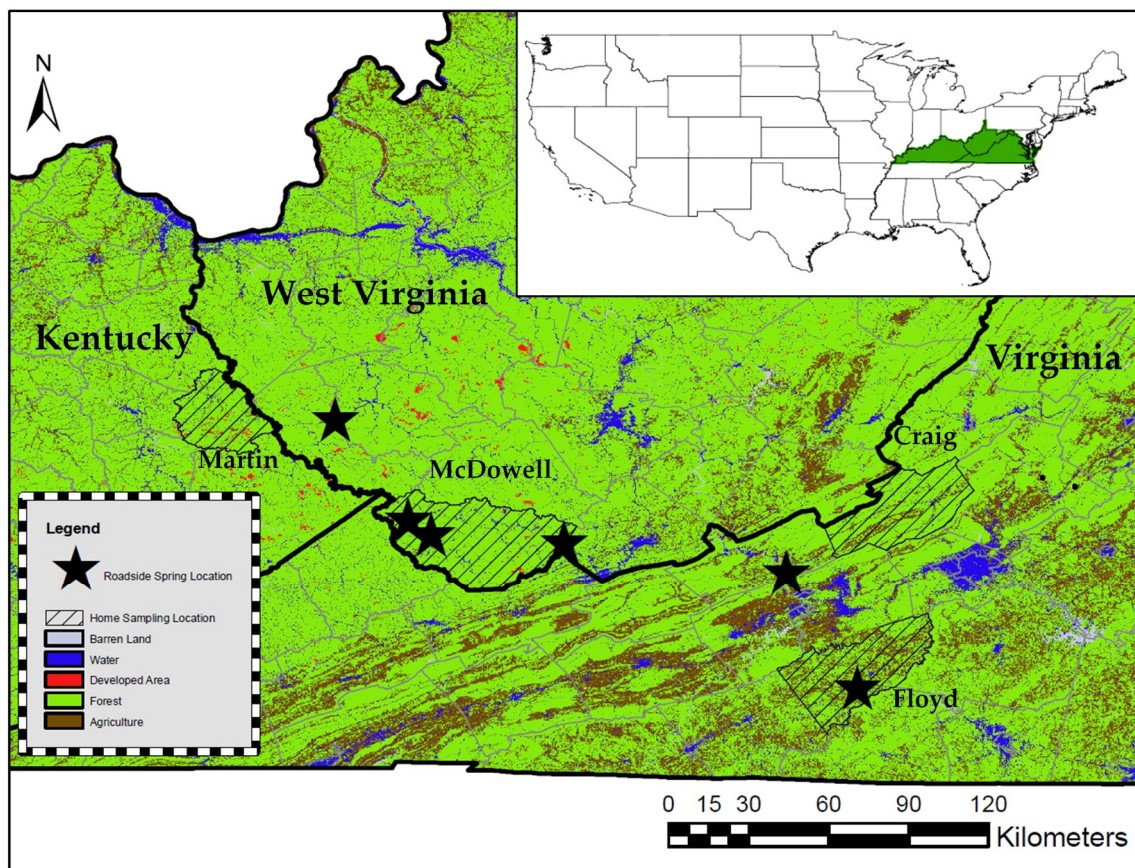

**Figure 1.** Map featuring the Central Appalachian region in relation to the United States, and the locations of roadside springs and household sampling.

## 2.2. Water Sample Collection and Surveying

The day prior to sampling, each resident received a home sampling kit including sampling instructions and four bottles for sample collection. Residents were instructed to collect their water samples from their most frequented POU in their house (e.g., the kitchen tap) and asked if they had any questions regarding the instructions. Per USEPA home sampling guidelines (EPA 815-F-18-022) [28], residents were instructed to leave their water stagnant in their pipes overnight and to obtain a first-draw sample in the morning, flush the pipes for 30 s, and then collect the second-draw sample, followed by two samples for nutrient and bacteriological analysis.

First- and second-draw samples were collected in pre-prepared acid-washed 250 mL bottles for metal cation analysis. A third acid-washed 250 mL bottle was used to collect a sample for anion analysis. The fourth sample was collected in a 100 mL sterile bottle for bacteriological analysis. After sample collection, residents were instructed to tightly cap the bottles, which were picked up the same day by the research team. Samples were packed in iced coolers for transport back to the laboratory and immediate bacteriological analysis.

The sampling kit also included a household survey (see Supplementary Materials, Figure S1), consisting of 18 multiple-choice and short answer questions designed to explore perceived water quality, water use, household plumbing characteristics, and alternative drinking water sources. Responses were coded in Microsoft Access and matched to home water quality results.

Roadside spring samples were collected immediately following collection of the household sampling kits to provide a comparison of water quality on the same sampling day. In total, seven spring samples were collected from the six springs during seven separate sampling events (i.e., one spring in McDowell County, West Virginia sampled on two occasions). Water samples were obtained

from the spring using one autoclaved 2 L polypropylene sampling bottle for bacteriological testing, and two 250 mL sampling bottles for metals and nutrients analysis. Additionally, water quality parameters including pH, specific conductivity, dissolved oxygen, and water temperature were obtained using a YSI Quattro Pro on-site (YSI Inc., Yellow Springs, OH).

### 2.3. Water Sample Analysis

Bacteriological analysis of water samples collected from homes and springs was completed for total coliform and *E. coli* via the Colilert defined substrate method immediately upon returning to the laboratory on the day of sampling (www.idexx.com, Westbrook, MN). Metals analysis for both home and spring samples was completed using the ICP-IMS process as stated in Standard Methods 3030D and 3125B [29]; the suite of metals included Na, Mg, Al, Si, P, S, Cl, K, Ca, Ti, V, Cr, Fe, Mn, Co, Ni, Cu, Zn, As, Se, Sr, Mo, Ag, Cd, Sn, Ba, Pb, and U. Standard Methods 4500-NH was used to assess $NO_3^-$ concentrations in the water samples and Standard Methods 300.0 was used to assess F concentrations [29]. Contaminant concentrations in first- and second-draw home tap water samples were compared in order to determine the maximum concentration detected for each of the analyzed constituents. This comparison was done in order to better determine the maximum health risk (i.e., worst-case scenario) a resident might be exposed to at their tap on the given sampling day.

After home and spring water samples were analyzed, study participants were contacted and provided water quality reports containing data for both their home tap water and their local roadside spring. In addition, a written letter was provided to each participant outlining any contaminant concentrations in home or spring samples that exceeded SDWA recommendations and their associated health risks. Also included in the letter were brief recommendations regarding water quality improvements for home and spring water.

### 2.4. Statistical Analysis

Statistical analysis of the differences between home and spring water quality was conducted via the Wilcoxon Rank Sum Test for non-parametric data. All tests were conducted in RStudio version 3.5.1 (RStudio, Boston, MA). After visually checking for normality using the "ggplot" data package, the Wilcoxon Rank Sum Test was run on each individual constituent analyzed, with significance defined at an alpha of 0.05.

## 3. Results and Discussion

### 3.1. Motivations for Collecting Spring Water

A total of 23 survey responses were collected from the 24 homes sampled. One individual did not have running water in their home, instead relying on hauling water from a private spring on their property or the local public spring, and therefore did not wish to complete the survey. It should be noted that while a total of 23 individuals returned completed surveys, some individuals chose not to answer certain survey questions. The majority of household samples (58.3%) were collected from homes connected to a municipal water system, 37.5% of the household samples were sourced from homes on private well water, and 4.2% were from homes connected to private springs (Table 1).

**Table 1.** Distribution of homes on private and municipal drinking water treatment systems in each community (n = 24).

| Community Location | # of Homes Using Private Wells/Spring Box | # of Homes Using Municipal Water Service |
|---|---|---|
| McDowell County, WV | 9 | 1 |
| Martin County, KY | 0 | 12 |
| Floyd County, VA | 0 | 1 |
| Craig County, VA | 1 | 0 |

The majority of survey respondents (82.6%) indicated that they do not trust their in-home POU tap water (Figure 2). This is not necessarily a surprising statistic given that this population was selected for its preference for spring water and desire for formal water quality testing. Study participants who did not trust their tap water were asked to specify the reasons behind their responses. When grouping these responses by common themes, one key driver for this mistrust appeared to be aesthetic issues: 42.1% of individuals cited issues of taste, smell, color, and/or odor as reasons that they did not trust their tap water. In addition, 21.1% of individuals cited a distrust of various facets of their public drinking water system (e.g., concerns with distribution line location or treatment system age); these responses were grouped into a "Distrust of public system" category. Individuals also cited health concerns (e.g., the recent death of a relative) and/or a general distrust of their tap water quality (e.g., water is "no good") as reasons they did not trust their drinking water; these responses were grouped into "Concerns about health effects" and "General distrust of water quality" categories.

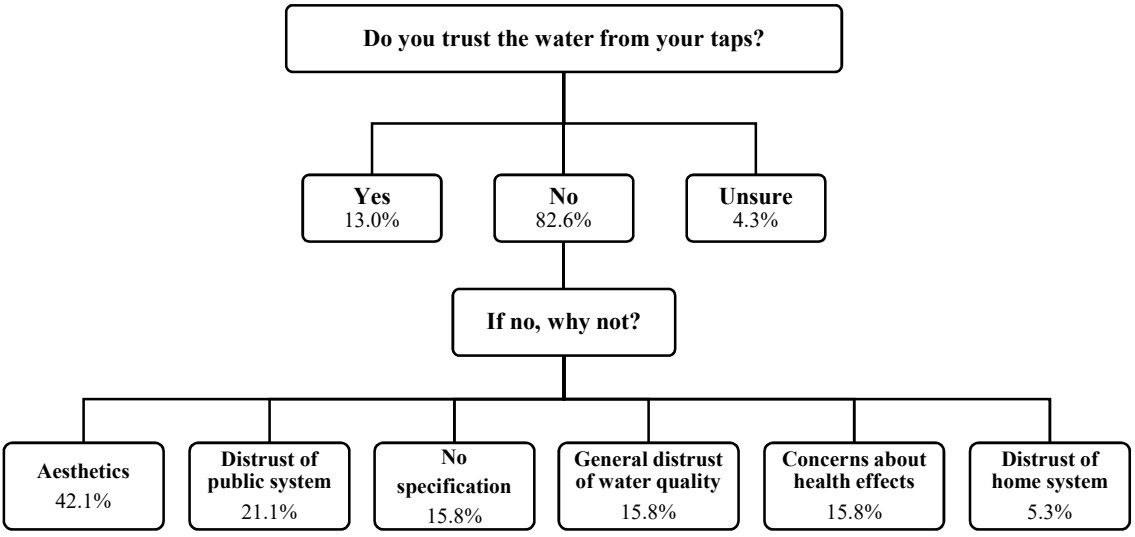

**Figure 2.** Study participant survey responses to questions regarding trust of in-home point-of-use tap water, categorized by common themes.

Study participants' distrust of their home tap water was evident in survey responses to questions regarding what participants use their tap water for day-to-day. Less than half of survey respondents reported that they were drinking their tap water (40%). The majority of respondents reported using their tap water for personal hygiene, such as bathing (95.7%) and brushing their teeth (73.9%), and for household tasks, such as cleaning (95.7%) and cooking (60.7%).

### 3.2. Trust in Roadside Spring Water

Only nine survey respondents explicitly responded that they were visiting roadside springs to obtain drinking water, with the majority leaving these questions blank (unanswered). This may reflect participant recruitment by neighbors for water quality testing and/or a desire to maintain privacy. The majority of these respondents visited their preferred spring at least once per month, and over half visited once per week to collect drinking water. A total of 44.4% of specified spring users live between three and eight kilometers away from the spring they visit, 22.2% live between eight and 16 km away, and only one respondent lives within two kilometers of the spring they visit. One survey respondent in particular drove an hour away from their home to collect roadside spring water for drinking.

The majority of spring survey respondents use the water they collect for drinking, cooking, brushing teeth, and providing water to their livestock and pets. The most common responses addressing why these individuals collect water from roadside springs as opposed to using their tap water fell into the category of "general distrust of home water". While the majority of total survey respondents (82.6%)

reported that they did not trust their home tap water, the majority of spring user respondents (62.5%) reported that they had no concerns relating to the quality of spring water that they collect. The distance that individuals travel to collect spring water, the impression that spring water is of superior quality compared to home water, and general distrust of home tap water, suggests that in this study group, an individual's perception of water quality impacts the drinking water source that they utilize.

### 3.3. Microbial Contaminants in Home and Spring Water Samples

Total coliform and *E. coli* are regulated as MCLs in the SDWA as they are considered indicators of pathogen presence [20]. A total of seven spring samples and 21 of the 24 home samples collected were analyzed for bacteriological contaminants (see Supplementary Materials, Table S1). All (100%) roadside spring samples tested positive for total coliform (Figure 3); the concentration in one sample from spring #4 in McDowell County, West Virginia exceeded the maximum detection limit of 2420 MPN/100 mL. Over half of the spring samples (57.1%) also tested positive for *E. coli* (Figure 3), a more serious indicator of potential infectious risk, with spring #4 yielding the highest concentration of 489 MPN/100 mL.

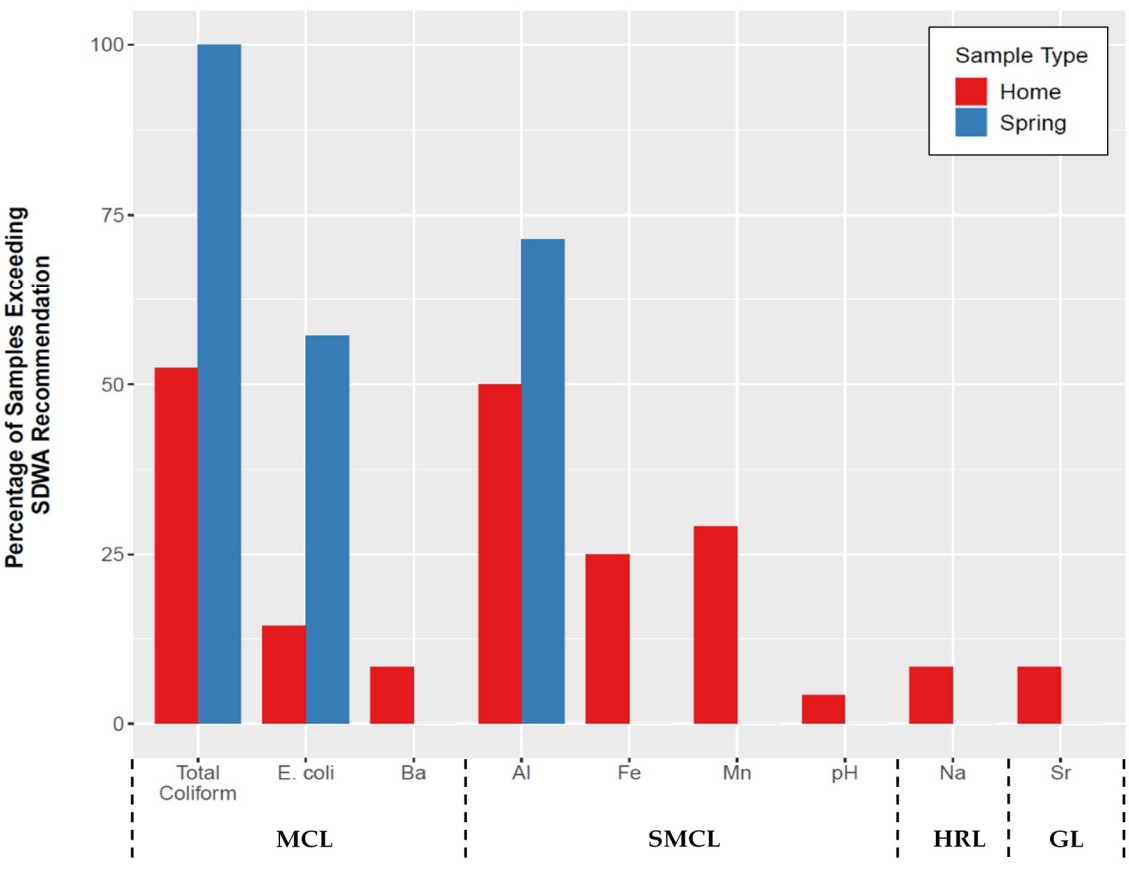

**Figure 3.** Percentage of home (n = 24) and spring (n = 7) samples exceeding Safe Drinking Water Act (SDWA) contaminant recommendations for detected constituents. For home samples, this data represents the highest concentrations detected between first- and second-draw samples. (MCL = Maximum Contaminant Level, SMCL = Secondary Maximum Contaminant Level, HRL = Health Reference Level, GL = Guidance Level).

Roughly half (52.4%) of all home samples analyzed tested positive for total coliform and 14.3% tested positive for *E. coli* (Figure 3). Bacteriological contamination was most prevalent in tap water samples collected from homes in McDowell County, West Virginia with observed concentrations of

*E. coli* as high as 461 MPN/100 mL. When analyzing the data based on water source, the private well/spring home tap water samples had substantially more total coliform (90% positive) and *E. coli* (30% positive) violations, compared to 18.2% and 0%, respectively, from home samples sourced from municipal systems. The higher presence of microbial contaminants in home samples from private sources is unsurprising, given that previous surveys have reported that microbial contamination can occur in private water supplies, though the incidence of contamination observed in this study is notably higher than typically observed [30]. The presence of microbial contaminants in home samples from municipal systems is more surprising as it is expected that residual chlorine from the water treatment process would combat the presence of total coliform in these samples.

While coliforms are present in the environment naturally [31], the presence of *E. coli* in over half of the spring samples and several home water samples indicates that the water is contaminated with human and/or animal fecal matter [32]. Pathogens associated with the fecal-oral route of transmission can cause gastrointestinal illness or distress to those who are exposed, including diarrhea, cramps, nausea, headaches, and other symptoms. Due to the associated symptoms, these pathogens may pose a heightened risk for infants, young children, the elderly, and anyone with compromised immune systems [33]. The presence of total coliform and *E. coli* in spring samples suggests that spring-users may face a risk of infection. The three home tap water samples that tested positive for *E. coli* were collected at homes using private well water, suggesting that homeowners may need added disinfection treatment in order to lower their risk of exposure to microbial contaminants.

The detection of total coliform and *E. coli* in spring samples in this study is consistent with results from Swistock et al. [27] who sampled 37 springs in Pennsylvania and reported near universal total coliform contamination, with a third of spring samples also positive for *E. coli*. These results also align with those of Krometis et al. [6], which reported total coliform and *E. coli* in 99% and 86% of 83 samples, respectively, from roadside spout springs in five Central Appalachian states [6].

*3.4. Metal Cation Contaminants in Home and Spring Water Samples*

Concentrations of Ba in two samples collected from McDowell County, West Virginia households exceeded the health-based MCL (Figure 3). No other home samples yielded constituents with concentrations that exceeded the health-based regulatory standards dictated in the SDWA. However, several samples had concentrations of nuisance contaminants (Al, Mn, Fe, Na, Sr) that exceeded unenforceable SMCL recommendations for aesthetic contaminants, and/or GL/HRL recommendations (Figure 3; see Supplementary Materials, Table S2).

Barium is regulated as an MCL by SDWA as it can increase blood pressure in exposed individuals, posing a health risk [33]. Increased blood pressure is of particular concern in the Appalachian region where known health disparities, such as incidence of heart disease, exist [14,34,35]. The two homes that exceeded the Ba MCL with concentrations of 5.7 ppm and 12.4 ppm, also exceeded the USEPA's GL for Na of 20,000 ppb with concentrations of 244,316 ppb and 604,435 ppb, respectively. These home tap water samples were both sourced from private wells which is perhaps unsurprising as Ba and Na can both occur naturally in deposits [22,33]. The GL for Na was developed for individuals on low-sodium diets of 500 mg/day [22]. High concentrations of Na, coupled with high concentrations of Ba present in the drinking water at these homes may be a concern for the residents as both excess Na and Ba can increase blood pressure, leading to larger health risks over time [22,33].

Increased levels of aluminum were detected in both home and spring water samples. While Al is currently included in the SDWA as an SMCL, some epidemiological research has suggested that Al levels above 100 ppb may promote the onset and progression of Alzheimer's disease and/or damage to cerebral function [36–40]. However, it is critical to note that both the USEPA and World Health Organization (WHO) have concerns regarding the validity of reported correlations between waterborne Al and cognitive function [41]. The majority of home samples that tested above the SDWA SMCL for Al were located in Martin County, Kentucky. These homes are all on the same surface water sourced municipal water system. This may suggest that the water treatment technique implemented at the

water treatment plant involves the use of Al which can lead to elevated levels in drinking water [41]. While Al in drinking water is not currently considered to be a health risk to consumers, the presence of the metal in home tap water can cause discoloration, dissuading residents from consuming the water for fear that it is unsafe.

Just over half of home samples from McDowell County, West Virginia exceeded Fe and Mn SMCLs. Elevated Fe and Mn can cause aesthetic issues in drinking water, including sedimentation, metallic taste, and staining of plumbing fixtures [42], which might lead consumers to seek alternative water sources [43]. The homes in McDowell County that had high levels of Mn and Fe obtained tap water from private wells. Constituents such as Fe and Mn often occur together in groundwater, particularly in deeper wells where natural sources are present and water has been in contact with rock for an extended period of time [43].

Tap water samples from two homes in McDowell County, West Virginia both on private well water, had Sr levels that exceeded HRL guidelines. Strontium can occur naturally in groundwater [44] and, though not yet included on the SDWA reference contaminant list, has been assigned an HRL of 1.5 mg/L by the USEPA [44]. This is because high levels of Sr, particularly when paired with poor nutrition, are thought to cause rickets [44]. Given that Appalachia is known to have a significant number of food deserts impacting diet, i.e., low-income census tracts where a significant number of households have low access to vehicular transportation and/or are more than 20 miles from their nearest supermarket [45,46], high Sr intake could pose a health concern.

### 3.5. Comparing Home and Spring Water Quality

A Wilcoxon Rank Sum Test was completed on water quality data to compare paired home and spring samples based on each metal cation, anion, and bacteriological constituent that was analyzed, with significance defined at 0.05 (see Supplementary Materials, Table S3). The difference between spring and home samples was statistically significant when comparing total coliform, Cd, F, $NO_3^-$, U, Cu, Pb, Ag, Al, Mn, $SO_4^{2-}$, Zn, Na, and Sr. Spring values had significantly higher concentrations of total coliform, U, Al, and $SO_4^{2-}$ while home samples had significantly higher concentrations of Cd, F, $NO_3^-$, Cu, Pb, Ag, Mn, Zn, Na, and Sr. While concentrations of the listed constituents were found to be significantly higher in either home or spring samples, it must be noted that of the 14 constituents, only four (total coliform, Al, Mn, Na) were found in home and spring samples at levels above SDWA recommendations. Of these four constituents, only total coliform is regulated by the SDWA as an MCL, posing a direct risk to human health. Al and Mn are SMCLs and Na is a HRL. Based on the differing nature of harmful constituents found in sampled household and spring water, residents may be exposed to different health risks (heavy metals or bacteriological) at different levels depending on their drinking water source selection.

Evidence collected on user perceptions in several previous studies in Central Appalachia [6,13,14,23–25], suggested that, based on resident's distrust of their home POU tap water, home samples would have significantly higher concentrations of metals, in particular, aesthetic contaminants such as Fe, Mn, and Al that can alter user experience. Based on several studies of roadside spring water quality in the Appalachian region [6,27], it was anticipated that roadside spring water samples would have higher concentrations of the microbial contaminants total coliform and *E. coli*, as these have been previously detected in springs across the region. The results of this study generally mirrored these expectations as home samples yielded higher concentrations of metals (Cu, Pb, Ag, Mn, Zn) although Mn was the only metal detected in home samples with concentrations above SDWA recommendations. Spring samples indeed yielded a significantly higher concentration of the microbial contaminant total coliform, however, it was expected that *E. coli* concentrations would also be significantly higher in roadside springs, which was not the case.

*3.6. Comparing Water Quality & Survey Responses*

Observations of generally higher levels of contaminants of aesthetic concern in household samples (Fe, Mn, Al) echo participants' perceptions of household water quality (Figure 4). When surveyed, participant's cited issues with the aesthetics of their tap water as the primary reason they did not trust it, stating, for example, that their tap water "doesn't taste right, has an odor, and is often discolored", has a "bad smell—like frog pond water with a commensurate taste", and "changes color sometimes it's clear sometimes it's milky looking sometimes it's gray". Understanding the impact of perception on source selection is critical when designing drinking water interventions. The faith that these respondents have in the quality of water from roadside springs is reflected not only in the time that they spend collecting water, but also the distance that they are willing to travel in order to do so. When asked why participants chose to collect roadside spring water, they provided several reasons relating both to spring and home water quality, stating that the spring water "doesn't have (the) chemical put in public water, natural source, better quality than comes from faucet" and that they "have bad water" and were "worried about drinking and cooking using my well water". Reports that these springs may be contaminated with fecal indicators, contaminants that do not change the taste or appearance of water, may not change water collection or usage behavior. For example, a study on alternative drinking water source selection by Delaire et al. [47] in West Bengal, India, found that awareness and dissatisfaction with water quality was most commonly related to the presence of tangible water contaminants such as the aesthetic contaminant Fe, and less commonly related to the presence of the tasteless, odorless, but highly toxic contaminant as.

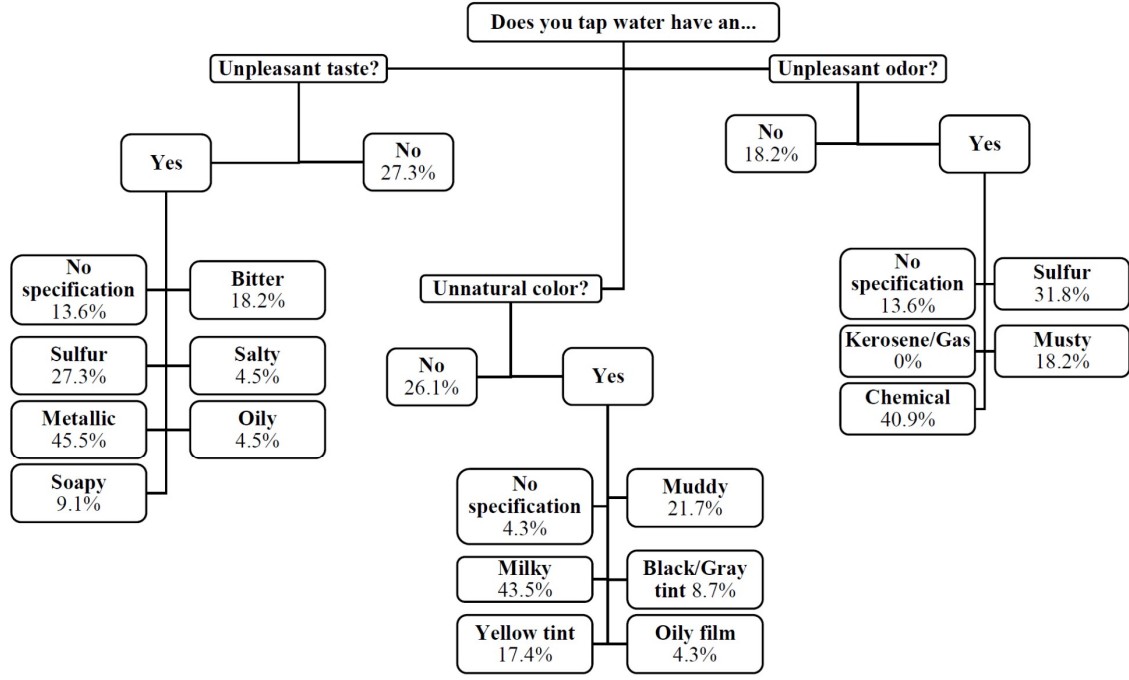

**Figure 4.** Survey responses to questions regarding in-home point of use (POU) tap water aesthetics.

*3.7. Study Limitations*

It is critical to note that water quality in environmental samples (i.e., surface or groundwater) can be extremely variable, and the values presented in this study represent only a single day's observation, both at the home tap and at roadside springs. Significantly more data, including rates of consumption, pathogenic profiles, and metals bioavailability, would be required to truly quantify the relative risks associated with the consumption of spring versus in-home POU water in any one of these households. However, in general, given the difference in significant constituents at each water source, it appears

that risks associated with the consumption of home tap water would largely be associated with concentrations of metals, whereas bacteriological exposures would be of greater concern following consumption of spring water.

A study limitation regarding the number of study participants who answered survey questions regarding spring use specifically (n = 9) compared to the total number of survey participants (n = 23) is also critical to note. The authors believe that the number of study participants answering questions about roadside springs may have been limited due to concerns about the repercussions of admitting spring use, including the condemning of a roadside spring site and subsequent loss of a water source that is perceived to be of high quality and may hold cultural significance to users. Additionally, as this was a community-based study, community partners were used in some locations to help recruit participants and it is possible that those recruited through this pathway may not regularly rely on roadside springs, despite living in a community where some residents are spring-dependent. While the number of participants explicitly reporting personal spring use is lower than the total number of survey participants, the authors believe this qualitative insight into the perception of spring water quality is still noteworthy and may help direct future research efforts in the Central Appalachian region.

## 4. Conclusions

Despite the availability of POU drinking water in most homes, some residents of Central Appalachia still choose to use alternative drinking water sources. This choice appears motivated by several factors. Forty-one percent of study participants cited poor perception of their in-home tap water quality stemming from unpleasant aesthetics, which can indeed be a useful indicator of substandard water quality. However, these individuals might be trading one set of water quality problems for another, since the roadside spring water they often rely on can contain elevated levels of microbial contaminants.

While the specific water infrastructure, access, and quality issues present in Central Appalachia may not directly transfer to other contexts, these results build upon and/or re-emphasize work on alternative drinking water sources and water quality perceptions from other nations. Most importantly, interventions aimed at improving water access and quality for any particular population should comprehensively consider consumer decision-making processes and factors. For example, though a new or upgraded centralized distribution system may appear favorable in some cases, such a solution may not address factors like cost, or perceptions based on legacy POU water quality problems or cultural norms. The selection of alternative drinking water sources is often based on the perception of water quality available at each source, with individuals often making decisions based on aesthetics or prior experience. In some cases, improvement of a preferred source (e.g., spring development or protection) and/or providing water quality information to consumers [48] may prove a more effective or accessible means of rebuilding trust in home tap water and reducing risks of adverse waterborne exposure than attempts to change behavior.

**Supplementary Materials:** The following are available online at http://www.mdpi.com/2073-4441/12/3/888/s1, Figure S1: Household survey distributed to study participants, Table S1: Distribution of biological contaminants detected across roadside spring and home tap water samples, Table S2: Distribution of inorganic ion contaminants detected across roadside spring and home tap water samples, Table S3: Results of Wilcoxon Rank Sum Test to determine statistical significance in comparison of paired home and spring water sample constituents.

**Author Contributions:** Conceptualization, L.-A.K. and E.S.; Data curation, H.P.; Formal analysis, H.P.; Funding acquisition, L.-A.K. and E.S.; Methodology, H.P., L.-A.K. and E.S.; Project administration, L.-A.K.; Supervision, L.-A.K.; Visualization, H.P., L.-A.K. and E.S.; Writing—original draft, H.P.; Writing—review & editing, L.-A.K. and E.S. All authors have read and agreed to manuscript publication.

**Funding:** This research has been funded by the Virginia Tech Institute for Critical Technology and Applied Science (ICTAS) (Grant #175453).

**Acknowledgments:** The authors would like to thank Laura Lehmann for field work preparation, lab maintenance, and sampling advice, Jeffrey Parks for assistance with inorganic ion analysis, Kelly Peeler for assistance with nutrient analysis, and Casey Schrading, Ethan Smith, and Austin Wozniak for assistance with sample collection and processing.

**Conflicts of Interest:** The authors declare no conflict of interest.

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
