# Peer review of "Springing for Safe Water: Drinking Water Quality and Source Selection in Central Appalachian Communities"

_water, doi:10.3390/w12030888_

Round 1

Reviewer 1 Report

General comments:

This mixed-methods source water quality and risk perception study is on an interesting topic addressing an underserved population within the Appalachian region of the United States. It appears to be well thought-out and draws helpful conclusions. I wondered about the time of day sampling took place (first- and second-draw), and temporal relationship with spring sampling (same day), and would suggest including these details earlier in the paper (e.g., in the abstract and/or aims). The approach was straightforward and logical, and presentation of the results could be strengthened in some ways to better showcase this (detail below). The text does not require major editing, but some language could be tightened (e.g., long sentences with excess small words and/or comma series). 

Specific comments:

  • Sarver affiliation superscript should be changed to “3”
  • There are some colloquial word choices and double negatives throughout (e.g., not impossible, not uncommon). Consider revising to more formal writing. The word “also” can often be deleted without altering meaning.

Abstract

  • Line 21: Consider clarifying whether POU samples included both spring water and piped water sources consumed at the home (I think it was just piped).
  • Line 24: Consider adding a quantitative measure (e.g., percentage) of those that did not trust the home tap water.
  • Line 25: Not sure why “in keeping with” - results don’t seem to agree (perhaps change to “Water quality results suggested…”)
  • Line 27: suggest “risks and perceptions” since both were measured by the study

Introduction

  • Line 35-36: There are many studies in many countries documenting use of alternative sources. I would add “for example” as a transition to introduce these select studies.
  • Line 49: Consider mentioning which region or states the Navajo Nation crosses.
  • Lines 49-53: Please break into two sentences.
  • Line 58: “national” and “for the United States” are repetitive, just need one
  • Line 60: upwards of -> more than
  • Lines 65-68: Suggest starting the next paragraph with “However…” (otherwise not clear how this statement is supported). An alternative would be to add some references here (e.g., Amjad et al. 2016. Water safety plans: Bridges and barriers to implementation in North Carolina).
  • Line 79: Don’t need to transition geography here since paragraph 3 already narrowed discussion to Appalachia. Suggest deleting “In keeping with observations from developing nations,”
  • Lines 87-88: “alter individual’s occupational behavior patterns” is unclear, consider rewording
  • Line 98: Please revise “Of those 12% that” (maybe missing punctuation?)
  • Line 100: efforts -> studies
  • Line 111: #3 missing open parenthesis
  • Line 113: reliance on

Materials and Methods

  • Line 118: two states -> three
  • I’d like a bit more detail about the springs… Did flushing apply to the spring samples? Did all springs have an enclosure/tap or were some open? If closed, did all have the same type of infrastructure? Are any managed, monitored or regulated?
  • For the statistics, it seems like a long list of metals were tested. Did you consider applying a familywise correction to the p-values?

Results and Discussion

  • Line 240: sentence should be revised (e.g., indicators of pathogen presence, correlated to disease risk)
  • Why not include Table S1 in the main text?
  • Line 263: infectious risk -> risk of infection
  • Line 265: bacteriological -> microbial contaminants (remember the E. coli is an indicator, not necessarily the pathogen)
  • Line 274: detail which type of samples (home or spring, etc.) had aesthetic contaminants
  • Table S2: Make sure to define every acronym so table is understandable apart from the text. Vertical spacing in the second column runs together (e.g., As > 10 ppb Ba…). Would it be possible to fit definitions on one line or create some separation? (same for table S3)
  • Line 284/303: Did you contact/follow up with these participants to explain the health risk? (please briefly report on the procedure)
  • Lines 286-295: For aluminum results, with all the caveats I’m not sure what the main message is here… just ignore findings or more temporal sampling needed? Maybe add a final sentence to wrap up the paragraph.
  • Please move Figure 3 just after the in-text reference
  • Section 3.5: would like some additional discussion of how the findings relate to your original hypothesis (e.g., expected/unexpected). Are some of these findings actually more related to surface vs. groundwater quality? Do we know about the source(s) of the tap water? Maybe mention how residual chlorine related to bacteria counts, and potential for contamination at the point of use/home piping.
  • In general, it seems like the quantitative results got more attention than the qualitative data (written responses on survey). If there was good qualitative data available, you could feature it by pulling key quotes into the discussion.
  • Line 342: this is an important point! consider adding some references to back it up (e.g., Delaire et al. 2017. Determinants of the use of alternatives to arsenic-contaminated shallow groundwater: an exploratory study in rural West Bengal, India)
  • Consider consolidating study limitations at the end of the discussion.

Conclusions

  • Consider including key values/quantitative measures to back up qualitative observations in first paragraph (e.g., commonly, often)
  • Line 351: bacterial -> microbial 
  • Line 353-354: this caveat downplays the similarities, I would cut or soften it

Reviewer 2 Report

The research question is an important one (why people with access to piped water turn to other alternatives and whether their perception of the water quality is unfounded). However, the study itself is only partially suitable for answering this question. 

The international studies cited in the opening paragraph seem to be arbitrarily chosen and not relevant to the topic of the paper. The study design is and the methods are appropriately chosen, though the study would have benefited from adding some further chemical elements to the analysis, such as ammonium and nitrite. It would have been also interesting to measure residual disinfectant in those sites, if any, where disinfection is applied. 

The most serious problem is the fact that only 9 respondents answered explicitly that they collect and drink spring water. It is unclear if the respondents use those springs, which were sampled, and if so, which one of them. This way it is difficult to assess the choices the residents make. 

Though first and second draw samples were both collected for the analysis of metals, the results of the two are not compared, though the first mainly gives information on the quality of the faucets (i.e. leaching of metals), and not that of the water itself. 

The conclusion that spring water is a higher microbial hazard while home tap water is more of a chemical hazard due to the presence of metals, is valid, but the health risk of the two are widely different. The observed E.coli levels in the springs (and in homes supplied with spring water) associate with fecal pollution posing an immediate risk of infection, while most of the detected metals have questionable health impact, and quite probably the concentration can be reduced by flushing the taps before use. For metals detected in tap water, the source of piped water (municipal or spring) is not analysed (or results are not presented. These aspects should be discussed, before the paper is reconsidered for acceptance. 

Reviewer 3 Report

Thank you for the opportunity to revise this interesting manuscript, which aims at understanding the perception (and distrust) of drinking water quality in Central Appalachian Communities.

I have found the study design appropriate. Results and Methods are clearly presented and discussed.

My only concern is that the authors did not mention how the quality of the POU water is communicated to the users. It seems that users do not have any official information about the quality of their home water, isn't it?

This aspect is key to the perception of drinking water quality, and it has been recently studied and demonstrated. In particular, when the communication is scarce, the risk perception reaches the highest level of distrust and the community outrage increases.

I suggest the authors add some information about the actual communication of the drinking water quality to the population, both in the introduction and the conclusion sections. In particular, the conclusion section must report some hypothesis of increasing the communication strategies, in order to manage the community outrage and distrust of drinking water quality.

I suggest to read the following paper that can be useful:

- Dettori, M.; et al. Population Distrust of Drinking Water Safety. Community Outrage Analysis, Prediction and Management. Int. J. Environ. Res. Public Health 2019, 16, 1004.

I hope the authors will follow the suggestion, as I believe that the study will be well received by the readers.

Round 2

Reviewer 1 Report

Well done! The authors incorporated my suggestions and made some other improvements as well. I look forward to seeing this paper published.

Reviewer 2 Report

Thank you for the amendments, the revised text gives a much clearer picture of the study findings. I still miss any reference to the impact of piping/faucet material for the in house water samples (comparison of first and second draw samples would easily reveal if any of the metals was leaching from the materials in contact with water), but otherwise I find the manuscript acceptable. 

Reviewer 3 Report

The authors have addressed all the issues raised.

I believe that the scientific quality of the manuscript has increased, and I can now recommend the publication on Water in the present form.